# Fungal Footprints: Soil Fungal Communities in Black Walnut and Red Oak Forests

**DOI:** 10.3390/microorganisms12112184

**Published:** 2024-10-30

**Authors:** Shaneka S. Lawson, Juan P. Frene, Niall D. Lue Sue

**Affiliations:** 1USDA Forest Service, Northern Research Station, Hardwood Tree Improvement and Regeneration Center (HTIRC), Department of Forestry and Natural Resources, Purdue University, 715 West State Street, West Lafayette, IN 47907, USA; ndluesue@gmail.com; 2School of Biosciences, University of Nottingham, Sutton Bonington, Nottingham LE12 5RD, UK; juanpablofrene@gmail.com

**Keywords:** fungal communities, hardwood plantations, *Juglans*, mycorrhizae, *Quercus*, soil properties

## Abstract

Soil fungal communities are critical for forest ecosystem functions in the Central Hardwood Region (CHR) of the USA. This evaluation, which took place in 2022–2023, investigates the influence of *Juglans nigra* (BW, black walnut) and *Quercus rubra* (NRO, Northern red oak) on soil properties and fungal community structures across three CHR sites. The objectives of this study are to investigate how the fungal communities identified beneath *J. nigra* and *Q. rubra* serve to influence biodiversity and soil health within hardwood plantations. Soils from two locations in Indiana and one in Michigan were examined and assessed for variations in fungal composition and diversity. Soil fungal communities were characterized using Illumina high-throughput sequencing while multivariate analysis was applied to analyze patterns in these fungal communities. These data provided insights into how environment, location, and tree species affect fungal community structure. Results indicate that *J. nigra* soils exhibited higher carbon (0.36%, 1.02%, 0.72%), nitrogen (25%, 29%, 56%), and pH (0.46, 1.08, 1.54) levels than *Q. rubra* soils across all three sites and foster greater fungal diversity. Specifically, *J. nigra* was associated with increased *Ascomycota* diversity, whereas *Q. rubra* supported a higher prevalence of *Basidiomycota*. *Basidiomycota* were negatively correlated with carbon and pH, while *Ascomycota* showed positive correlations with these variables. These findings highlight how crucial it is to understand how different tree species influence fungal communities and, consequently, how they influence forest soil health. Our findings serve to improve forest management practices by emphasizing the importance of fungal communities in maintaining the function and resilience of an ecosystem. Our study underscores that grasping these specific interactions is essential for effective forest management, especially when considering how to use fungal communities to boost plant growth. This work focuses on hardwood plantations rather than either agricultural ecosystems, monocultures, or native forests, thus filling a gap in the current literature where many studies are limited to specific fungal groups such as mycorrhizae. In future research, it is important to examine a wider range of tree species. This will deepen our understanding of fungal community dynamics and their impact on maintaining healthy forest ecosystems. Our hardwood plantation focus also notes the potential for adaptive forest management as environmental conditions change.

## 1. Introduction

The Central Hardwood Region (CHR) is a pivotal ecological zone, encompassing approximately 90% of the hardwood forests in the United States. This region spans eight states, including Missouri, Arkansas, Illinois, Tennessee, Indiana, Ohio, Kentucky, and Pennsylvania. It serves as a critical area for biodiversity, forest management, and ecological research due to its extensive hardwood forests that provide essential ecosystem services, support diverse flora and fauna, and contribute significantly to regional and national economies [1]. The CHR has characteristics of a temperate deciduous forest and is mainly dominated by deciduous species like oak-hickory, maple-basswood, bottomland hardwood, and the conifer representing less than 1% [2]. Many of the hardwood tree species including oaks, cherry, and walnut species, are considered highly valuable woods and can provide good long-term investment opportunities through timber production [3]. Wolz et al. [4] showed that black walnut plantations are an economically competitive alternative to row crops in the CHR. Additional benefits of hardwood forests and plantations are wildfire habitats, watershed protection, and wildlife conservation [3].

The sustainability and health of hardwood forest ecosystems rely on intricate interactions occurring within the soil. Fungal communities play a key role in these processes. These interactions are critical and further underscore the forests’ economic and ecological significance. Fungi are crucial components of soil ecosystems and play multifaceted roles in shaping ecosystem dynamics, maintaining soil health, and promoting the resilience of terrestrial ecosystems. Soil fungi are critical for proper nutrient cycling, plant growth, and overall ecosystem stability because of their diverse ecological roles [5,6,7]. This intricate web of fungal activities encompasses many various aspects of soil functioning, from the degradation of organic material to nutrient turnover, soil structure preservation, and the establishment of plant symbioses. In doing so, their indispensable contributions to ecosystem health and functioning are emphasized [8].

Fungi play specialized roles in forming symbiotic relationships with trees. Soil fungi can contribute to disease suppression in soils by provoking pathogenic microorganisms and priming the plant’s immune system response [9]. Some fungal species produce antibiotics and other bioactive compounds that inhibit the growth of soil-borne pathogens and reduce the risk of plant diseases while enhancing overall plant health and productivity [7]. Mycorrhizal fungi are particularly essential among the various functional groups of fungi in hardwood forests and are notable for their ability to induce systemic resistance in plants. This process improves the plants’ capacity to withstand pathogen attacks by triggering a broad-spectrum immune response [10].

Mycorrhizal fungi are essential to both natural and managed forest systems for their distinct and specialized roles in promoting tree growth and resilience. This cooperative symbiotic relationship is especially true in plantation settings where soil conditions may be suboptimal. Soil fungi are instrumental in the regeneration of forest ecosystems following disturbances such as logging, fire, or natural disasters [7]. Pioneer fungi colonize bare soil to facilitate the establishment of vegetation. These fungi break down organic matter to create a suitable substrate for seed germination and subsequent seedling growth [5,6]. Early successional fungi initiate the colonization process leading to the development of more complex fungal communities and the eventual restoration of diverse forest ecosystems. In plantation forestry, fungi have essential duties that are integral to the establishment, growth, and productivity of tree crops [11]. Mycorrhizal fungi improve the resilience to environmental stresses and enhance nutrient uptake and water absorption in tree seedlings [12]. Plantation managers are typically aware of this key function of mycorrhizae and often inoculate tree seedlings with mycorrhizal fungi to promote their growth and survival. This practice is especially beneficial in degraded or nutrient-poor soils [13,14].

Understanding the myriads of roles and functions for specific fungal communities associated with different tree species is crucial to the optimization of forest management practices. This knowledge is vital as it allows for the development of more targeted interventions that can enhance tree resilience, improve nutrient uptake, and ensure the sustained longevity of natural and managed forest ecosystems. It is important to know the local fungal species associated with each tree species to improve the development of better inoculum for hardwood trees in the CHR. This study aims to expand our understanding across a broader geographical range by building on previous research on fungal communities associated with *J. nigra* and *Q. rubra* in Michigan that indicated that microbiome composition differs significantly between these two tree species [15,16]. Therefore, this study examines the structure, composition, and diversity of fungal communities in hardwood plantations across three sites in Indiana and Michigan. We hypothesize that fungal communities identified in this study will differ by both plant species and geographic location. Other efforts in this area typically focus on agricultural or native ecosystems but our work addresses an understudied area in current studies by focusing on hardwood plantations. This study highlights the vital role fungi play in nutrient cycling, ecosystem stability, and soil health while uncovering intricate relationships among fungal diversity, soil traits, and tree species. The work offers practical applications for forest management while trees endure and adapt to changing environmental conditions and notes that understanding these variations could provide key insights for forest management strategies.

## 2. Materials and Methods

### 2.1. Site Characterization

This study investigated the forest plantation ecosystems of J. nigra and Q. rubra at three locations: Grand Rapids, MI (42.9634° N, 85.6681° W), West Lafayette, IN (40.4259° N, 86.9081° W), and Rushville, IN (39.6092° N, 85.4464° W). The plantations were established in Blount loam (silty clay loam or clay loam till) soil in 2008. Each location consisted of three 20 × 20 m blocks, with the plots spaced at least 50 m apart. In Grand Rapids, the study area included two adjacent blocks, while a nearby plant-free site (DB) served as a reference (Appendix A). The sites in West Lafayette were about 6 miles apart, separated by agricultural fields and small forest patches, while the Rushville sites were 130 miles apart, with various developments, fields, and forest patches in between. Slope gradients ranged from 1% to 3%, with some areas reaching up to 6%. Climate data indicated an average annual temperature of 9.5 °C and 932 mm of precipitation in Grand Rapids, MI. In Butlerville, IN, temperatures ranged from 22°F to 86°F, with approximately 49 inches (1245 mm) of precipitation annually and 14 inches of snowfall. Lafayette, IN showed temperatures from 17°F to 84°F, with annual precipitation of 41 inches (1041 mm) and 20 inches of snowfall. Precipitation patterns followed seasonal trends, with maximum rainfall in May and June. Climate data was sourced from U.S. Climate Data (https://www.usclimatedata.com).

### 2.2. Soil Sample Collection and DNA Amplification

Soil samples were gathered in November 2019 and in March, June, and September 2020. Sampling was designed to represent the overall variability of the research area and to account for differences in terrain and vegetation cover. From each zone, 25 cores (3 cm diameter) were extracted at depths of 0–10 cm and 10–20 cm across multiple locations to ensure heterogeneity in soil properties. Cores were collected evenly throughout each zone to ensure comprehensive representation of the area before being combined into composite samples after excluding leaf litter. Samples were placed in labeled polyethylene bags, kept on ice, and transported to the lab. After sieving through a 2 mm mesh, each composite was split into two, with one half stored at −20 °C before extraction of DNA with the DNeasy 96 PowerSoil Pro Kit (Qiagen, Germantown, MD, USA) and sequenced with a MiSeq 1000 (Illumina Inc., San Diego, CA, USA).

In this study, multiple genetic markers were employed to assess fungal communities from environmental DNA (eDNA) soil samples. The highly conserved 18S rRNA gene was amplified using the primer pair FR1 (5′-AICCATTCAATCGGTAIT-3′) and FF390 (5′-CGATAACGAACGAGACCT-3′), which is commonly used in broader taxonomic studies targeting higher taxonomic levels such as phylum or class. Additionally, the D1-D2 region of the 28S rRNA gene, which provides valuable phylogenetic information, was amplified using the primers NL1 (5′-GCATATCAATAAGCGGAGGAAAAG-3′) and NL4 (5′-GGTCCGTGTTTCAAGACGG-3′), particularly for fungal identification at the species level within Basidiomycota and Ascomycota. For more detailed fungal identification, we also targeted the highly variable internal transcribed spacer (ITS) regions. The ITS1 region was amplified using the fungal-specific primer ITS1F (5′-CTTGGTCATTTAGAGGAAGTAA-3′) paired with ITS2, while the ITS2 region was amplified using the primers ITS4 (5′-TCCTCCGCTTATTGATATGC-3′) and ITS86F (5′-GTGAATCATCGAATCTTTGAA-3′). These primers were selected due to their established utility in fungal biodiversity assessments, taxonomy, and phylogenetic studies.

DNA samples were amplified in triplicate in 25 µL PCR reactions containing 5 µL of 1× PCR buffer, 1 µL of 0.2 µM of each primer, 1.8 µL of 0.2 mM dNTPs, 1 µL of 1.5 mM MgCl_2_, and 0.2 µL of 1 U Taq polymerase (New England Biolabs, Ipswich, MA, USA), with the final volume adjusted using nuclease-free water. The amplification protocol consisted of denaturation for 25 cycles at 95 °C for 30 s, annealing at 55 °C for 30 s, and extension at 72 °C for 30 s with a final extension at 72 °C for 5 min. Purified amplicons were quantified and prepared for sequencing with the Illumina Nextera XT kit, generating 250 bp paired-end reads. DNA quantification was carried out using a Qubit 4 Fluorometer (Thermo Fisher Scientific, Waltham, MA, USA), and DNA quality was assessed with an Agilent 2100 Bioanalyzer (Agilent Technologies, Santa Clara, CA, USA), ensuring all samples adhered to the established integrity criteria.

### 2.3. Soil Physicochemical Analysis

The remaining half of the soil sample was air-dried, ground into a fine powder, and sieved through a 2 mm mesh to remove larger particles, rocks, pebbles, and other debris. This prepared sample was then shipped to the Soil Health Assessment Center in Columbia, MO, USA (https://soilhealth.missouri.edu/). The optimization of laser diffraction for agricultural applications was detailed by Caspers et al. [17], while Cheng et al. [18] adjusted it for use in assessing soil organic carbon distribution under various management practices. Soil pH was measured using advanced techniques to ensure accuracy and efficiency. Soil pH was quantified with automated sensors from Hanna Instruments in accordance with protocols outlined by Sparks and Page [19] for examining the effects of pH on microbial communities.

Soil organic carbon (SOC) and nitrogen were measured with high precision using the LECO TruMac Carbon/Nitrogen Analyzer and Elemental Vario Macro and LECO TruMac N analyzers. The LECO TruMac provided accurate SOC quantification through dry combustion, as demonstrated by Melkani et al. [20] in agricultural regions. Nitrogen content was determined with the Elemental Vario Macro and LECO TruMac N analyzers, which were utilized by Rytter and Rytter [21]. Phosphorus content in soil samples was measured using the Bray-1 extraction method [22]. A total of two grams of air-dried soil were shaken with 20 mL of 0.03 M ammonium fluoride and 0.025 M hydrochloric acid for 5 min, then filtered. Phosphorus in the filtrate was determined colorimetrically at 882 nm, following Murphy and Riley [23], and expressed as mg/kg soil.

### 2.4. Bioinformatics Analysis

We used the standardized workflow version 1.16 of the DADA2 pipeline. The DADA2 algorithm [24] was used to generate amplicon sequence variants (ASVs), a method that avoids the need for the traditional 97% sequence similarity clustering. We primarily used the default settings in the DADA2 pipeline, making a few specific adjustments to optimize the process. For both the forward and reverse fungal sequences, we set a minimum read length of 270 bp and limited the maximum expected error after truncation to three. Afterward, the forward and reverse sequences were merged, and chimeric sequences were removed using the consensus method. The final step involved aligning and classifying the fungal sequences against the UNITE v10 database [25] for accurate identification. For the ITS DNA metabarcoding libraries, high-throughput sequencing data sets were entered into the Sequence Read Archive (SRA) of the National Center for Biotechnology Information (NCBI) under BioProject number PRJNA1035935.

### 2.5. Statistical Analysis

The R software environment (version: 4.1.2, R Development Core Team, 2017) was utilized to get data interpretation and graphical representation. The *ggplot2* package (v3.5.0) in R was utilized to construct all of the figures [26]. Variations in the soil nutrient content and microbial community diversity (Shannon alpha-diversity index; α-diversity) were evaluated using ANOVA and Tukey’s post hoc test (*stats*, v4.3.0, and *agricolae*, v1.3-7, packages). Visualizing beta-diversity (β-diversity) patterns was made easier with the use of Non-Metric Dimensional Scaling (NMDS) ordinations based on Bray–Curtis dissimilarity matrices (ordinate function, *phyloseq* package, v1.46.0). Permutation-based analysis of variance (PERMANOVA) was used to evaluate the effects of the treatment on β-diversity. Multiple comparisons and Bonferroni adjustment were then used to ascertain any variations in community composition (adonis function, *vegan* package, v2.4-6). Using normalized ASV data, the Shannon diversity index was computed using the *phyloseq* package. Furthermore, the R scripts utilized for data analysis and processing are openly available at https://github.com/JuanFrene/CHR-Fungi.

## 3. Results

### 3.1. Soil Physical and Chemical Properties

The soil properties are described in Table 1. *J. nigra* presented significantly higher values of Corg, Nin, and pH than *Q. rubra*; however, there were no significant differences by location. The available *p* showed an interaction between plant species and location, *J. nigra* presented higher values of *p* in Michigan and South Indiana, while *Q. rubra* was higher in West Indiana (Table 1). Additionally, we analyzed the soil chemical profile using a multivariate analysis. The Principal Coordinate Analysis (PCoA) explained that the first principal components accounted for 70.16% and 19.47% of the PCoA1 and PCoA2 axis, respectively, of the total variation of the soil’s chemical properties. Samples were clustered by tree species (Figure 1). *J. nigra* was positively correlated with PCo1 and the Michigan and South Indiana samples clustered together. The *Q. rubra* samples were negatively correlated with PCoA1.

### 3.2. Soil Fungal Community Structure and Diversity

The PERMANOVA analysis of the fungal communities from all sites and plant species indicated that the communities were significantly structured by plant species, and the interaction between plant species and location, with explained variation ranging from 60% (for plant species) to 2% (for interaction). The location effect accounted for less than 1% of community variation (Figure 2a). The multivariate NMDS analysis showed that samples were grouped by plant species in the NMDS1 with *J. nigra* being on the negative axis while the samples from *Q. rubra* were along the positive NMDS1 axis (Figure 2b).

The *J. nigra* samples presented significantly higher values of alpha-diversity compared with *Q. rubra* in the three sites (*p* = 2.301 × 10^−5^). Additionally, the alpha-diversity decreased from West Indiana to South Indiana and Michigan (Figure 3). There was a positive association between Shannon’s alpha-diversity index with N available (r = 0.39, *p* = 0.031) and pH (r = 0.44, *p* = 0.013) (Table 2).

### 3.3. Soil Fungal Composition Associated with Tree Species

The fungal communities were composed mainly of *Basidiomycota* and *Ascomycota*, which represented almost 95% of the fungal species in the soil, followed by *Chytridiomycota*, *Mortierellomycota*, and *Glomeromycota* (Figure 4). The *Basidiomycota* phyla were significantly greater in the *Q. rubra* samples (*p* = 2.05 × 10^−6^), but the *Ascomycota* taxa were significantly higher in the *J. nigra* samples (*p* = 1 × 10^−6^). *Basidiomycota* phyla showed a significant and negative correlation with Corg (r = −0.49, *p* = 0.005), Ning (r = −0.41, *p* = 0.023), and pH (r = −0.51, *p* = 0.003), while the *Ascomycota* showed a positive correlation with Corg (r = 0.52, *p* = 0.002), Ning (r = 0.42, *p* = 0.017), and pH (r = 0.55, *p* = 0.001) (Table 2).

Following the analysis of the fungal phylum abundance, we analyzed the twenty most abundant fungal species (Figure 5). The genus *Russula* (four species, *Basidiomycota*) were enhanced in *Q. rubra* in the three sites; in addition, *Russula farinipes* was significantly higher in West Michigan. Other species enhanced in *Q. rubra* were *Tuber lindaslei* (*Ascomycota*), *Tomentella tenuirhizomorpha* (*Basidiomycota*), and *Cortinarius saniosus* (*Basidiomycota*). On the contrary, the genus *Archaeorhizomyces* (three species, *Ascomycota*) and *Trichoderma* (two species, *Ascomycota*), both *Ascomycota*, were enhanced in *J. nigra*, with *Archaeorhizomyces sborealis* being significantly higher in this plant species in West Indiana (Figure 5). Other fungal species enhanced in *J. nigra* were *Talaromyces rufus* (*Ascomycota*), *Coniochaeta verticillata* (*Ascomycota*), *Atrocalyx nordicus* (*Ascomycota*), and *Woswasia atropurpurea* (*Ascomycota*).

## 4. Discussion

This research offers important perspectives on the diversity and makeup of soil fungal communities connected to various hardwood tree species across varied locations in the Central Hardwood Region (CHR). The findings offer a nuanced understanding of the intricate relationships between soil properties, fungal community structure, and tree species, enhancing our comprehension of these interactions within forest ecosystems.

### 4.1. Soil Properties and Tree Species

Our analysis of soil physical and chemical properties highlights notable differences between the tree species studied. Specifically, *Juglans nigra* (black walnut) exhibited significantly higher values of carbon (Corg), nitrogen (Nin), and pH compared to *Quercus rubra* (red oak). Bernreiter and Teijeiro [27] found that different tree species can significantly influence soil Corg and Nin in temperate forests and that variations in nutrient composition also affect nutrient cycling and forest health. Rhoades and Stokes [28] investigated how tree species composition is linked to soil dynamics. Their work indicated that different tree species led to significantly different quantities of nutrients available for uptake. These findings substantiate our hypothesis that distinct tree species like *J. nigra* and *Q. rubra* influence soil chemistry which in turn impacts fungal community dynamics. The interaction between plant species and location was particularly evident in available phosphorus (*p*), where *J. nigra* showed higher values in Michigan and South Indiana, while *Q. rubra* had elevated levels in West Indiana. Such variations in soil nutrients across locations underscore the localized effects of tree species on soil chemistry. The Principal Coordinates Analysis (PCoA) of soil chemical properties, accounting for 70.16% and 19.47% of the variance in the first two principal components, revealed clustering patterns by tree species. PCoA helps visualize the variation in soil chemistry between *J. nigra* and *Q. rubra*. In this study, PCoA1 and PCoA2 explain most of the variation and show that *J. nigra* soils, especially in Michigan and South Indiana, have higher Corg, Nin, and pH levels. The *Q. rubra* soils, particularly in West Indiana, have lower values. The clear clustering of samples by tree species indicates that soil chemistry is highly likely to be influenced by tree type rather than location. The PCoA findings align with the broader research results and show how different tree species drive soil characteristics. These data reinforce the connection between tree species and soil properties. *J. nigra* samples were positively correlated with PCoA1 and clustered together for Michigan and South Indiana sites, indicating a consistent chemical signature associated with this species. Conversely, *Q. rubra* samples were negatively correlated with PCoA1, suggesting different chemical influences in these soils. These findings imply that soil properties associated with specific tree species may significantly influence fungal community structure and distribution.

### 4.2. Fungal Community Structure and Alpha-Diversity

The analysis of fungal community structure, using Permutational Multivariate Analysis of Variance (PERMANOVA), demonstrated that fungal communities were significantly structured by plant species and the interaction between plant species and location, with plant species explaining 60% of the variation and location accounting for less than 1%. PERMANOVA reaffirms our visual findings with statistical evidence. It shows that tree species account for 60% of the variation in fungal community structure and that geographic location has very little impact. In fact, location explains less than 1% of the variation, and evaluation of the interaction between species and location only adds 2%. This analysis clearly confirms that tree species is the key factor driving differences in fungal communities. PERMANOVA strengthens the conclusion that *J. nigra* and *Q. rubra* shape distinct fungal ecosystems by quantifying these influences and noting their statistical significance. The Non-Metric Multidimensional Scaling (NMDS) results obtained here corroborate this finding and show that fungal communities from *J. nigra* and *Q. rubra* are separated along the NMDS1 axis. This separation highlights the impact of tree species on fungal community composition and emphasizes that distinct fungal communities are associated with each species. Our NMDS analysis in this study clearly shows how different fungal communities are associated with both J. nigra and Q. rubra. The groups cluster on opposing sides of the NMDS plot, with J. nigra on the negative axis and Q. rubra on the positive. This separation emphasizes that each tree species supports its unique fungal community. NMDS makes it easier to see these distinct patterns and offers a straightforward way to interpret the complex relationships in the data. Alpha-diversity analyses further revealed that *J. nigra* supports significantly higher fungal diversity compared to *Q. rubra* across all sites. Peay et al. [29] explored how fungal communities changed over space and time in temperate forests and indicated that different tree species led to the variation in fungal community diversity and structure. The increase in alpha-diversity in *J. nigra* soils could be linked to its higher levels of Corg and pH, which were positively associated with fungal diversity indices. The observed decrease in alpha-diversity from West Indiana to South Indiana and Michigan reflects potential environmental gradients or changes in soil conditions across these locations.

### 4.3. Dominant Fungal Phyla and Composition

Our findings on fungal composition indicate that *Basidiomycota* and *Ascomycota* dominate the fungal communities, constituting nearly 95% of the fungal species observed. *Basidiomycota* were significantly more prevalent in *Q. rubra* soils, whereas *Ascomycota* were more abundant in *J. nigra* soils. The negative correlation of *Basidiomycota* with Corg, nitrogen, and pH, contrasted with the positive correlation of *Ascomycota* with these variables, suggests distinct ecological roles and preferences of these phyla concerning soil properties. This aligns with the observations of Tedersoo et al. [8], who also noted the dominance of these phyla in soil fungal communities.

### 4.4. Species-Level Composition and Geographic Variations

The analysis of the twenty most abundant fungal species revealed notable trends. *Russula* spp. (*Basidiomycota*) were more common in *Q. rubra* soils, with *Russula farinipes* being significantly higher in West Michigan. In contrast, *Ascomycota* genera such as *Archaeorhizomyces* and *Trichoderma* were more prevalent in *J. nigra* soils, with *Archaeorhizomyces sborealis* showing significant increases in West Indiana. These findings highlight species-specific associations and geographic variations in fungal community composition. The dominance of *Russula* in *Q. rubra* soils is consistent with previous studies [30,31], where *Russula* has been identified as an ectomycorrhizal genus associated with a wide range of tree species. Similarly, the presence of *Archaeorhizomyces* in *J. nigra* soils aligns with the observations of Baba and Hirose [32], who reported this genus to be prevalent in forest soils. Additionally, Bononi et al. [33] reported that phosphorus-solubilizing *Trichoderma* spp. from Amazon soils enhanced plant growth. These results parallel findings from our CHR study that showed that fungal communities foster nutrient cycling and ecosystem health. Similarly, Rao et al. [34] highlight how *Trichoderma atroviride* can suppress disease while promoting growth, reinforcing the CHR research’s emphasis on tree species-specific fungal interactions that enhance forest management practices for plant growth promotion. Additionally, the role of *Trichoderma* as a plant growth promoter and biocontrol agent, highlighted by Rao et al. [34], Stewart and Hill [35], Baldrian and Valaskova [36], and Tyśkiewicz et al. [37], suggests potential applications for forest management practices.

### 4.5. Alternative Research Perspectives

The findings presented here suggest that tree species have a significant impact on soil fungal communities. However, other researchers have reported outcomes that differ from the results obtained in this study. Gao et al. [38] noted that soil microbial communities are linked to soil texture and soil moisture levels in addition to tree species composition. Likulunga et al. [39] stated that tree species and land use type influenced soil fungal community composition. While Gacura et al. [40] agreed that tree species influenced fungal community richness, they proclaimed that nutrient status and ecosystem type was unexpectantly discovered to be more of a key factor. Boeraeve et al [41] and Beule et al. and [42] wrote that location was of greater significance than species alone. They also indicated that tree species and environmental factors interact in complex ways to influence fungal community composition. This disparity of thought and the results between research groups highlights the complexity of soil fungal communities and reinforces the need for continued investigations into this interconnected ecosystem network.

### 4.6. Implications for Forest Management

Understanding the specific fungal communities associated with different tree species has practical implications for forest management and restoration strategies. The significant variation in fungal diversity and composition based on tree species and location underscores the importance of considering fungal communities in forest management practices. Bödeker and Tedersoo [43] and Xue et al. [44] did not emphatically imply that tree species alone significantly affect soil microbial communities but highlighted the critical role of soil conditions and management practices. Targeted interventions, such as species-specific inoculations or soil amendments, could enhance tree resilience and productivity by leveraging the beneficial roles of mycorrhizal and other soil fungi. Hickman and Atherton [45] noted that mycorrhizal fungi were vital to forest restoration and suggested that the incorporation of fungi could improve tree health. Singavarapu et al. [46] indicated that a higher tree species diversity led to more diverse and functional fungal communities while Pereira et al. [47] showed that mixed-species systems contribute to healthier microbial communities and more effective nutrient cycling processes in forest ecosystems. These results concur with this study in reinforcing the importance of tree species composition and fungal diversity for the improvement of ecosystem functioning. This study adds to our knowledge of soil fungal communities within the CHR, revealing complex interactions between tree species, soil properties, and fungal species richness. Future research ventures should continue exploring these relationships in various additional tree species and across a plethora of other geographic locations. This will further refine forest management strategies and take steps towards improving ecosystem sustainability.

### 4.7. Broader Ecological Context

The relationships observed between fungal phyla and soil properties in our present study mirror those of El-Baky and Amara [9], who described similar interactions between soil chemistry and fungal community structure. Ostensibly, the continued observation of significantly higher abundances of *Basidiomycota* in *Q. rubra* and *Ascomycota* in *J. nigra* indicates that these phyla likely have functional roles in the varied soil environments. These fungi also have differing decomposition strategies. *Basidiomycota* is more effective at breaking down recalcitrant compounds like lignin [37,48] while *Ascomycota* appears to favor labile carbon sources [49], which further reinforces their ecological roles.

It is important to note that these fungal interactions extend beyond decomposition. Tree-symbiotic fungi such as mycorrhizae are essential for plant nutrient uptake and growth [37]. Also, in previous studies, tree species have been shown to drive soil biodiversity. Some researchers have shown that the dominant tree species in a forest also influence soil microbial communities [37]. The fungal genera identified in this study, including *Russula* and *Archaeorhizomyces*, have broad ecological relevance. In particular, *Russula* is ectomycorrhizal and associated with over 500 tree species [30,31], and *Archaeorhizomyces* plays a critical role in soil nutrient cycling [32,50,51]. Thus, this research study contributes to our understanding and provides additional details regarding the intricate relationships between tree species, soil properties, and fungal communities. We aim to improve upon the current knowledge available regarding forest ecology. Our work will also highlight future research directions while providing background information and practical strategies for enhancing forest health and sustainability by comparing our findings with previous research [52,53,54].

Overall, this study has a few key limitations to consider. We might be overlooking small-scale differences in fungal diversity that could reveal important details about local conditions by combining multiple soil cores into composite samples. In addition, since the research was limited to sites in Indiana and Michigan, the results might not apply to other areas or hardwood plantations nationwide that are exposed to different environments. Seasonal changes in fungal communities, limitations in identifying fungi through sequencing methods, and not accounting for interactions with other microorganisms like archaea may hinder our ability to obtain a complete picture of the ecosystem.

## 5. Conclusions

Our findings indicate that tree species have a substantial influence on soil fungal community structure and diversity. The dominant fungal phyla were noted to be *Ascomycota* and *Basidiomycota*. We found that tree species accounted for 60% of fungal community richness. Less than 1% was linked to location regardless of longitude. Distinctively, *J. nigra* was associated with a copiotrophic fungal community that was primarily composed of *Ascomycota* while *Q. rubra* supported a more oligotrophic fungal community dominated by *Basidiomycota*. These specific fungal lifestyle strategies were closely correlated with soil chemical parameters and showed a positive association with *Ascomycota* and a negative one with *Basidiomycota*. In addition, significant differences in fungal alpha-diversity were observed. A higher fungal diversity was observed in *J. nigra* compared to *Q. rubra* (*p* = 2.301 × 10^−5^), with markedly higher diversity being reported at the West Indiana site. Several fungal taxa with potential as plant growth promoters were identified here, which presents opportunities for future forestry applications.

This research contributed substantially to enhancing our understanding of the complex relationships between soil fungal communities, tree species, and soil chemistry within the Central Hardwood Region (CHR). Notably, *J. nigra* soils exhibited higher levels of carbon, nitrogen, and pH, which corresponded to increased fungal diversity compared to *Q. rubra*. Principal Coordinates Analysis (PCoA) and Non-Metric Multidimensional Scaling (NMDS) analyses supported the notion that tree species strongly influence fungal community composition. In fact, distinct fungal communities were supported by *J. nigra* and *Q. rubra*. Our results reaffirm the dominance of *Basidiomycota* in *Q. rubra* soils, which was negatively correlated to Corg and pH. *Ascomycota* was dominant in *J. nigra* soils and positively correlated with these traits. The results presented here highlight the unique ecological roles of these fungal phyla in relation to soil properties. These findings align with many previous studies and underscore the importance of tree species-specific fungal communities in forest ecosystems.

Forest managers can implement practical strategies to manage fungal communities specific to different tree species based on findings from this study. *J. nigra* trees support a high-nutrient fungal community and its beneficial relationship with *Ascomycota* fungi is essential. Additional nutrient enhancement may come in the form of introduced mycorrhizal fungi native to the area. Conversely, the *Basidiomycota*-dominated communities beneath *Q. rubra* require much lower nutrient levels and thrive best in nutrient-limited areas. Thus, using organic mulches and limiting fertilization may spur organic matter decomposition by *Basidiomycota* and result in an improved soil structure. Despite the allelopathic effects of juglone exuded by *J. nigra* roots, a mixed-species planting with both *J. nigra* and *Q. rubra* can thrive. The selection of juglone-tolerant *Q. rubra* varieties can bolster beneficial fungal relationships and may result in improved overall forest productivity and soil health. Consistent soil monitoring (pH, nutrients) is vital and manual applications of lime may balance soil conditions and support desired fungal richness. Long-term fungal community monitoring in areas of varied land management practices may contribute additional understanding of these intricate relationships over time, and collaborative efforts with scientists in various associated science fields (e.g., ecology, entomology, geology, mycology, organic chemistry) can provide input on the creation of a comprehensive forest management and monitoring program considering future variations in climate. An exploration of these strategies can benefit current plantations while continuing research efforts steadily refine proposed practices.

The implications of these study results for forest management are significant and suggest that fungal communities tailored to specific tree species should be incorporated into management and restoration strategies. Future research should expand to include a broader range of tree species and geographic locations, delve into the functional roles of individual fungal species, and assess the effects of forest management practices on fungal community dynamics. Overall, the information provided here enhances our understanding of forest ecosystems and provides practical insights for promoting forest health and longevity.

## Figures and Tables

**Figure 1 microorganisms-12-02184-f001:**
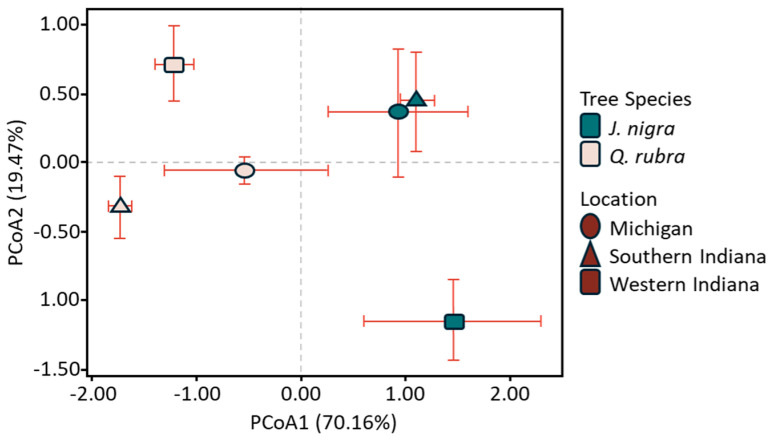
A Principal Coordinate Analysis (PCoA) for the three study sites based on the following standard soil properties: Corg, organic carbon; Ning, inorganic nitrogen; Pav, available phosphorus; and pH.

**Figure 2 microorganisms-12-02184-f002:**
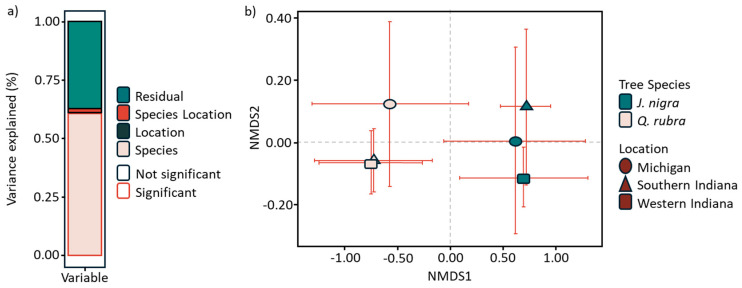
(**a**) A PERMANOVA analysis showing differences in fungal community structure; (**b**) A Non-Metric Multidimensional Scaling (NMDS) plot illustrating the distribution of fungal communities for *J. nigra* and *Q. rubra*.

**Figure 3 microorganisms-12-02184-f003:**
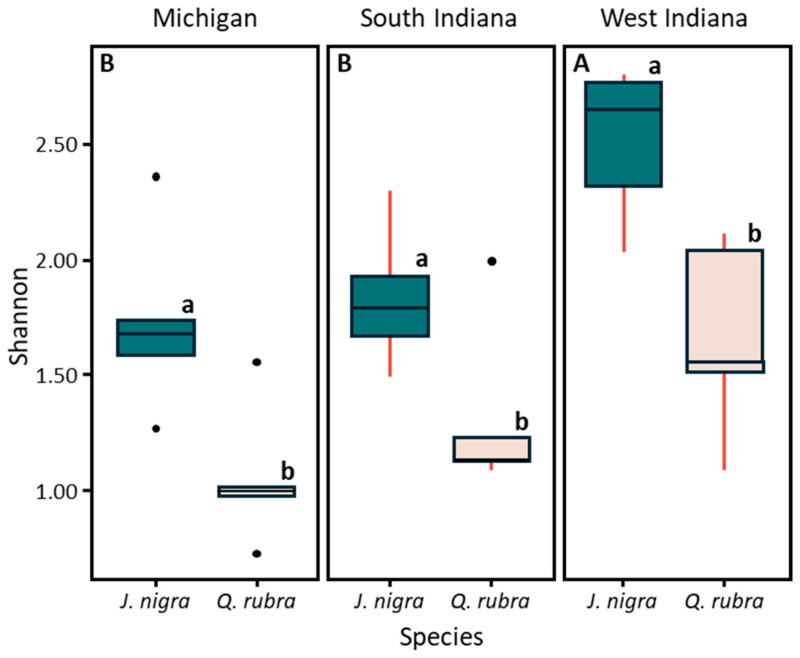
The changes in Shannon alpha-diversity by site and tree species in the Central Hardwood Region (CHR). Boxplot denotes mean ± SD (standard deviation). Different lowercase letters are significantly different (*p* < 0.05) among the different tree species (capital letters) and sites (upper letters) (ANOVA with Tukey’s honest significant difference (HSD)).

**Figure 4 microorganisms-12-02184-f004:**
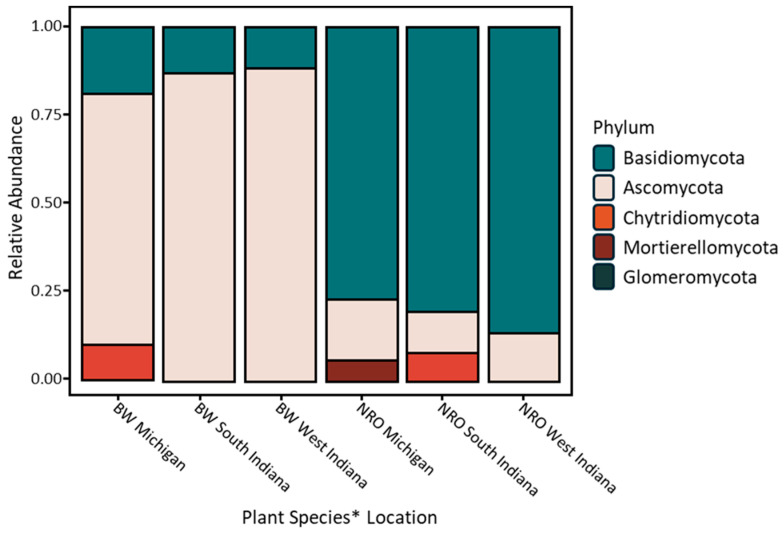
A compositional bar plot of the relative abundance of fungal phyla in soil samples across sites. *J. nigra* = BW, black walnut; *Q. rubra* = NRO, Northern red oak.

**Figure 5 microorganisms-12-02184-f005:**
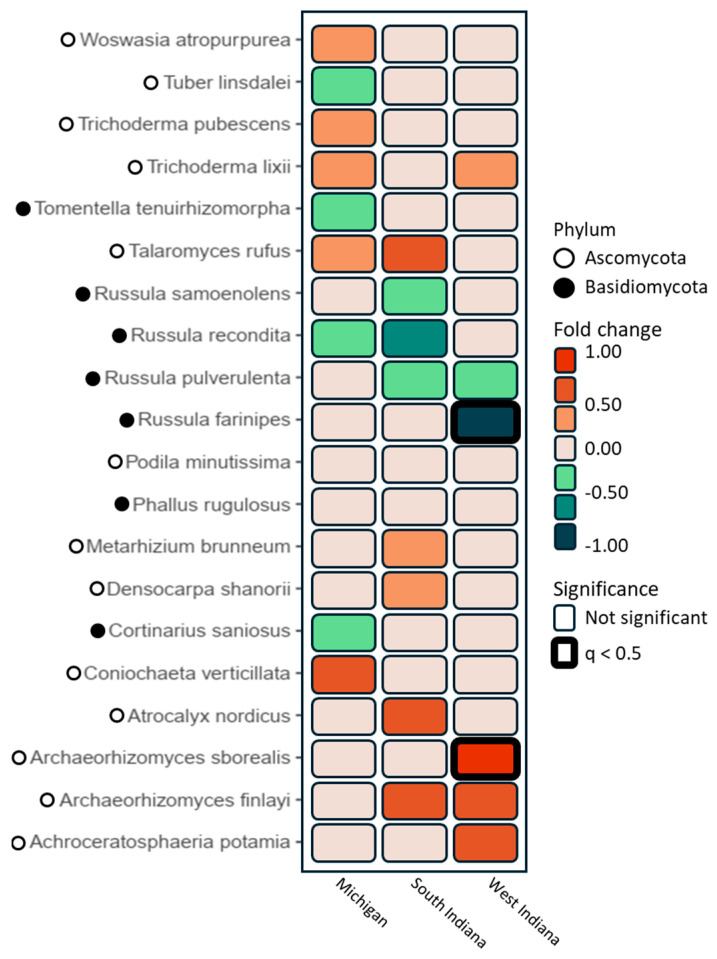
A heatmap showing differential abundance between *J. nigra* and *Q. rubra* for the twenty most abundant fungal species. Species were identified as belonging to *Ascomycota* and *Basidiomycota*.

**Table 1 microorganisms-12-02184-t001:** The basic soil properties at the three study locations.

	Species	Corg (%)	Ning	Pav	pH
Michigan	*J. nigra*	1.84	12.1	26.9	6.46
	*Q. rubra*	1.48	9.04	16.7	6
South Indiana	*J. nigra*	2.1	10.3	26.6	6.64
	*Q. rubra*	1.08	7.36	9.86	5.56
West Indiana	*J. nigra*	2	15.3	15.4	7.04
	*Q. rubra*	1.28	6.76	21	5.5

Corg = Soil organic carbon; Ning = Inorganic Nitrogen; Pav = Available P.

**Table 2 microorganisms-12-02184-t002:** The Pearson correlation between Shannon index and fungal phyla with soil properties.

Parameter	Corg (%)	Nin	Pav	pH
Shannon index	0.26	0.39 *	−0.02	0.44 *
*Basidiomycota*	−0.49 **	−0.49 *	−0.18	−0.51 **
*Ascomycota*	0.52 **	0.42 **	0.2	0.55 ***
*Chytridiomycota*	−0.15	−0.31	−0.22	−0.3
*Mortierellomycota*	−0.30	0.06	−0.14	−0.18
*Glomeromycota*	0.2	0.06	0.01	0.09

Corg = Soil organic carbon; Ning = Inorganic Nitrogen; Pa = Available P, * = *p* < 0.05, ** = *p* < 0.01,*** = *p* < 0.001.

## Data Availability

The Sequence Read Archive (SRA) of the National Center for Biotechnology Information (NCBI) has all raw sequencing data stored under the BioProject number PRJNA1035935. Additionally, the R scripts that were used to process and analyze the data are available at https://github.com/JuanFrene/CHR-Fungi.

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
