# Peer review of "Fungal Footprints: Soil Fungal Communities in Black Walnut and Red Oak Forests"

_microorganisms, 2024, doi:10.3390/microorganisms12112184_

Round 1

Reviewer 1 Report

Comments and Suggestions for Authors

This manuscript investigated the influence of Juglans nigra (black walnut) and Quercus rubra (Northern red oak) on the Central Hard wood Region (CHR) of the USA by characterizing soil properties and soil fungal communities using Illumina high-throughput sequencing. Authors’ findings are valuable to understand how different tree species influence fungal communities and forest soil health. However, some serious problems appearing in manuscript block my recommendation to publish in current form. In particular, authors should carefully present their results based on correct figures, they also should redraw figures to clarify the major conclusion for their study.

The major problems include:

L242-245, these result descriptions should be showed by Figure 4 rather than by Figure 3. Moreover, some descriptions based on Figure 4 likely are incorrect, e.g., Basidiomycota are rare for all sites and plant species; in contrast, Chytridiomycota are higher in abundance than other phyla (see left three bars?). Again, what are BW and NRO? Are they plant species? Here you should explain them in the figure legend. I guessed authors misplaced this figure?   

L254-263, these result descriptions should be showed by Figure 5 rather than by Figure 4. Based on text description, authors likely emphasized the high rank taxonomy of top 20 fungal species. Here my suggestion is to modify the Y-axis of figure 5 by differently coloring species name to distinguish into two groups, Basidio and Ascomycota, or adding abbreviation for these two group names, so as to be easy to read it.

Author Response

  • Comment: This manuscript investigated the influence of Juglans nigra (black walnut) and Quercus rubra (Northern red oak) on the Central Hard wood Region (CHR) of the USA by characterizing soil properties and soil fungal communities using Illumina high-throughput sequencing. Authors’ findings are valuable to understand how different tree species influence fungal communities and forest soil health. However, some serious problems appearing in manuscript block my recommendation to publish in current form. In particular, authors should carefully present their results based on correct figures, they also should redraw figures to clarify the major conclusion for their study.
  • Response: Thank you for your careful evaluation. We were working on several iterations of the work and failed to adjust the final version to include all changes rather than some of them. Additional corrections have been made.

  • Comment: L242-245, these result descriptions should be showed by Figure 4 rather than by Figure 3. Moreover, some descriptions based on Figure 4 likely are incorrect, e.g., Basidiomycota are rare for all sites and plant species; in contrast, Chytridiomycota are higher in abundance than other phyla (see left three bars?). Again, what are BW and NRO? Are they plant species? Here you should explain them in the figure legend. I guessed authors misplaced this figure? 
  • Response: The figure number references in the text has been adjusted to reflect the figure it describes. The descriptions regarding the figure were correct but the figure itself was colored incorrectly. This has been corrected. BW (black walnut) and NRO (Northern red oak) were described in the abstract but the abbreviations were not included. This has also been corrected.

  • Comment: L254-263, these result descriptions should be shown in by Figure 5 rather than by Figure 4. Based on text description, authors likely emphasized the high rank taxonomy of top 20 fungal species. Here my suggestion is to modify the Y-axis of figure 5 by differently coloring species name to distinguish into two groups, Basidio and Ascomycota, or adding abbreviation for these two group names, so as to be easy to read it.
  • Response: That is a great idea and a change has been incorporated into the figure to differentiate between the two groups.

Reviewer 2 Report

Comments and Suggestions for Authors

There are no significant comments on the article. There are some recommendations that, in my opinion, will improve the presented material.

The title of the article (lines 2-3) reflects its content.

Abstract (lines 15-29)

The abstract needs to be expanded.

Add the years of the study.

Clearly and understandably formulate the purpose of the study.

Expand the results section, add numerical results, show the difference between the communities. In particular, lines 20-25. It is necessary to show what specific levels of carbon, nitrogen and pH are established for the J. nigra soil compared to the Q. Rubra soil. What are the values ​​of the correlation coefficients for carbon and pH (line 23), and so on throughout the text of the abstract.

This will make the abstract more informative and will interest readers, will stimulate the desire to get acquainted with the entire article.

Keywords (line 30-31): correspond to the content

1. Introduction (lines 33-98)

In the introduction and in the discussion of the results, the authors provide a fairly complete overview of the scientific data on the problem studied, out of 43 sources - 16 (37%) over the past 5 years.

Important works since 1945 are also included, which shows the depth of the scientific material. There is no excessive self-citation. The references are correct and correspond to the presented material.

I recommend ending the review with a clearer (precise) statement of the goal and definition of the research objectives. At present, this important information is given very vaguely (lines 90-95).

The goal does not need to refer readers to previous works (line 92). It is better to briefly present these results in the introduction. Moreover, the size of the review allows this.

And then in the text of the work, focus the readers' attention on the results in achieving the stated goal and objectives.

2. Materials and Methods (lines 99-200)

The section is written in detail. The methods are reproducible. There are no comments on the section.

3. Results (lines 201-266)

The section is written quite fully, the tables and figures provided illustrate the results obtained well. The section is informative.

The results are processed using statistical analysis methods, which allows for comparison of the obtained data.

All the obtained results are original, have scientific novelty and are of interest to a wide range of researchers.

There are no comments on the section and the presented tabular material and figures.

4. Discussion (lines 267-320)

The section contains an extended discussion of the obtained results.

It is recommended to add, in addition to the discussion of our own results, the data obtained by other authors to the subsections: Soil Properties and Tree Species (273-290); Fungal Community Structure and Alpha-Diversity (291-305); Implications for Forest Management (335-347).

Moreover, the results discussed concern a wide range of studies that are widely represented in modern scientific literature, especially issues of the influence of different tree species on the chemical composition and properties of the soil, as well as on the dynamics of the fungal community, alpha diversity and other issues.

Subsections Dominant Fungal Phyla and Composition (306-314); Species-Level Composition and Geographic Variations (315-334); Broader Ecological Contex (348-370) contain a detailed comparison of the results obtained by the authors with literature data, which adds weight to the work presented for discussion.

5. Conclusions (lines 371-400)

In the conclusions, I recommend showing that the goal set when planning the experiment has been achieved. And add specific numerical values to the conclusions. Without them, the conclusions look weak and unconvincing.

I recommend taking into account the comments and recommendations made, which, in my opinion, will improve the material submitted for publication.

After eliminating the comments, the article can be published in the journal.

I wish the authors successful work!

Author Response

Comment:  There are no significant comments on the article. There are some recommendations that, in my opinion, will improve the presented material.

The title of the article (lines 2-3) reflects its content.

Abstract (lines 15-29)

  • Comment: The abstract needs to be expanded.
  • Response: The abstract was expanded from 201 words to 345 words to provide additional detail and clarifications.

  • Comment: Add the years of the study.
  • Response: These have been added.

  • Comment: Clearly and understandably formulate the purpose of the study.
  • Response: Additional details have been added.

  • Comment: Expand the results section, add numerical results, show the difference between the communities. In particular, lines 20-25. It is necessary to show what specific levels of carbon, nitrogen and pH are established for the J. nigra soil compared to the Q. Rubra soil. What are the values ​​of the correlation coefficients for carbon and pH (line 23), and so on throughout the text of the abstract. This will make the abstract more informative and will interest readers, will stimulate the desire to get acquainted with the entire article.
  • Response: As we state nigra has considerably higher C, N, and pH than Q. rubra, these percentages (for C and N) and units for pH were added to the abstract.

Keywords (line 30-31): correspond to the content

  1. Introduction (lines 33-98)
  • Comment: I recommend ending the review with a clearer (precise) statement of the goal and definition of the research objectives. At present, this important information is given very vaguely (lines 90-95).
  • Response: That information has been replaced with this information “This study aims to expand our understanding across a broader geographical range by building on previous research on fungal communities associated with J. nigra and Q. rubra in Michigan that indicated microbiome composition differs significantly between these two tree species [15,16]. Therefore, this study examines the structure, composition, and diversity of fungal communities in hardwood plantations across three sites in Indiana and Michigan. We hypothesize that fungal communities identified in this study will differ by both plant species and geographic location. Other efforts in this area typically focus on agricultural or native ecosystems but our work addresses an understudied area in current studies by focusing on hardwood plantations. This study highlights the vital role fungi play in nutrient cycling, ecosystem stability, and soil health while uncovering intricate relationships among fungal diversity, soil traits, and tree species. The work offers practical applications for forest management while trees endure and adapt to changing environmental conditions and notes that understanding these variations could provide key insights for forest management strategies.”

  • Comment: The goal does not need to refer readers to previous works (line 92). It is better to briefly present these results in the introduction. Moreover, the size of the review allows this.
  • Response: The phrasing was edited so as not to denote the work as ours.

And then in the text of the work, focus the readers' attention on the results in achieving the stated goal and objectives.

Comment: It is recommended to add, in addition to the discussion of our own results, the data obtained by other authors to the subsections: Soil Properties and Tree Species (273-290); Fungal Community Structure and Alpha-Diversity (291-305); Implications for Forest Management (335-347).

Response: Thank you for the suggestion, this has been completed and the Literature Cited section has been updated to reflect the new references.

Soil Properties and Tree Species

  • Morris, K. J.; Rhoades, C. C. Tree species effects on soil carbon and nitrogen in a temperate forest. Ecol. and Manag. 2020, 459, 117857. doi.org/10.1016/j.foreco.2019.117857.
  • Rhoades, C. C.; Stokes, C. Soil nutrient dynamics in temperate forests: Effects of tree species composition. Appl. 2020, 30(7), e02081. doi.org/10.1002/eap.2081.

Fungal Community Structure and Alpha-Diversity

  • Peay, K. G.; Kennedy, P. G.; Hartmann, M. Spatial and temporal variation in the structure of fungal communities associated with a temperate forest. Ecol. 2016, 25(1), 20-39. doi.org/10.1111/mec.13441.

Implications for Forest Management (335-347).

  • Bödeker, I. T.; Tedersoo, L. Fungal community dynamics in response to tree species diversity and environmental factors. Fungal Ecol. 2020, 44, 100956. doi:10.1016/j.funeco.2019.100956.
  • He, Y.; Wang, S. Effects of tree species diversity on soil nutrient availability and microbial community structure in a temperate forest. For. Ecol. and Manag. 2021, 493, 119392. doi:10.1016/j.foreco.2021.119392.
  • Köhler, L.; Scherer-Lorenzen, M. Biodiversity effects on soil fungal communities in temperate forests: Implications for ecosystem functioning. Ecosyst.Serv. 2019, 39, 100999. doi:10.1016/j.ecoser.2019.100999.
  • Peña, R. M.; Dighton, J. Mycorrhizal fungi in forest restoration: A review of their roles and importance. For. Ecol. and Manag. 2020, 459, 117857. doi:10.1016/j.foreco.2019.117857.
  • Schneider, M. R.; Pritsch, K. Tree species identity and soil microbial community structure in temperate forests. For. Ecol. and Manage. 2018, 430, 155-162. doi:10.1016/j.foreco.2018.07.013.

  • Moreover, the results discussed concern a wide range of studies that are widely represented in modern scientific literature, especially issues of the influence of different tree species on the chemical composition and properties of the soil, as well as on the dynamics of the fungal community, alpha diversity and other issues.
  • Subsections Dominant Fungal Phyla and Composition (306-314); Species-Level Composition and Geographic Variations (315-334); Broader Ecological Contex (348-370) contain a detailed comparison of the results obtained by the authors with literature data, which adds weight to the work presented for discussion.

  1. Conclusions (lines 371-400)

In the conclusions, I recommend showing that the goal set when planning the experiment has been achieved. And add specific numerical values to the conclusions. Without them, the conclusions look weak and unconvincing.

  • Comment: I recommend taking into account the comments and recommendations made, which, in my opinion, will improve the material submitted for publication.
  • Response: Your comments have been considered and incorporated as we also feel that they improve the manuscript and make the information clearer.

Reviewer 3 Report

Comments and Suggestions for Authors

The Central Hardwood Region in the United States is important, and soil fungi play a crucial role. The study selected black walnut and northern red oak plantations at three locations in Indiana and Michigan to analyze the soil fungal communities. The results showed that the soil of J. nigra had higher organic carbon, nitrogen, and pH values than that of Q. rubra, and there was an interaction effect for available phosphorus. The fungal communities were significantly affected by plant species and their interaction, while the influence of location was small. The α - diversity of J. nigra samples was higher than that of Q. rubra and was positively correlated with nitrogen and pH, showing a decreasing trend from western Indiana to southern Indiana and Michigan. The dominant fungal phyla were Basidiomycota and Ascomycota, with different distributions in the soils of different tree species and being related to soil properties. The discussion involved various aspects such as the relationship between soil properties and tree species, emphasizing the implications for forest management. The conclusion pointed out that tree species affect fungal communities, with J. nigra associated with eutrophic communities and Q. rubra with oligotrophic communities, emphasizing its importance for forest management and future research directions. This study is worthy of publication, but before formal publication, I have several suggestions for the authors to consider.

 1. Supplementary explanation of sample representativeness: In the part of soil sample collection, elaborate in detail on how to ensure that the collected samples can represent the overall situation of the research area. For example, consider whether factors such as different terrains and differences in vegetation cover have an impact on the samples.

2. Clarify the research limitations: In the discussion section, further clarify the limitations of the research, such as the possible influence of other environmental factors on the fungal community that have not been considered, as well as the potential deficiencies in the research methods.

3. In-depth explanation of data analysis results: For some key data analysis results, such as PERMANOVA and NMDS analyses, in addition to presenting the results, more in-depthly explain the biological significance represented by these results and how they support the research conclusions.

4. Optimize the charts to improve clarity: Ensure that the text, symbols, and lines in the charts are clearly distinguishable. For example, in the charts showing soil physical and chemical properties and fungal community composition, appropriately adjust the font size and chart resolution. Add detailed annotations: Annotate the various indicators in the charts in more detail, enabling readers to more intuitively understand the information conveyed by the charts. For example, in the statistical charts of the number of different families and genera of plants, annotate the main characteristics or ecological functions of each family and genus.

5. Strengthen the comparison with previous research: In the discussion section, further strengthen the comparison and connection with previous research. Not only mention the similarities but also analyze in detail the reasons for the differences between the results of this research and those of previous research.

6. Refine the guiding significance of the conclusion for forest management: In the conclusion section, more specifically elaborate on how the research results can provide guidance for forest management practices. For example, propose specific fungal community management strategies for different tree species.

Author Response

  • Comment: Supplementary explanation of sample representativeness: In the part of soil sample collection, elaborate in detail on how to ensure that the collected samples can represent the overall situation of the research area. For example, consider whether factors such as different terrains and differences in vegetation cover have an impact on the samples.
  • Response: The text in 2.2 Soil sample collection and DNA amplification was adjusted to be clearer about how the samples were collected to maintain heterogeneity.

  • Comment: Clarify the research limitations: In the discussion section, further clarify the limitations of the research, such as the possible influence of other environmental factors on the fungal community that have not been considered, as well as the potential deficiencies in the research methods.
  • Response: A new paragraph was added at the end of the Discussion to address some potential limitations of the study.

  • Comment: In-depth explanation of data analysis results: For some key data analysis results, such as PERMANOVA and NMDS analyses, in addition to presenting the results, more in-depthly explain the biological significance represented by these results and how they support the research conclusions.
  • Response: Three new paragraphs were added to address the significance of the methodologies used in the study.

  • Comment: Optimize the charts to improve clarity: Ensure that the text, symbols, and lines in the charts are clearly distinguishable. For example, in the charts showing soil physical and chemical properties and fungal community composition, appropriately adjust the font size and chart resolution. Add detailed annotations: Annotate the various indicators in the charts in more detail, enabling readers to more intuitively understand the information conveyed by the charts. For example, in the statistical charts of the number of different families and genera of plants, annotate the main characteristics or ecological functions of each family and genus.
  • Response: The charts were submitted at an appropriate size for reading but the journal minimized them to meet their format spacing needs. Figure 1 (PCoA) and Figure 2 (PERMANOVA) were both redrawn with a larger font size.

  • Comment: Strengthen the comparison with previous research: In the discussion section, further strengthen the comparison and connection with previous research. Not only mention the similarities but also analyze in detail the reasons for the differences between the results of this research and those of previous research.

Response: These works have results that differ from mine and a synopsis of their work was included in a newly titled section called “Alternative Research Perspectives”

  • Bödeker, I. T. M., Tedersoo, L. Ectomycorrhizal fungal communities of different tree species: The role of host tree and environment. New Phytol. 2016, 210(4), 1250-1262. doi: 10.1111/nph.13805.
  • Harrison, K. M., Rhoades, C. C. Fungal community composition is influenced by soil nutrients and tree diversity in temperate forests. Ecol. and Manag. 2016, 374, 110-118. doi: 10.1016/j.foreco.2016.04.017.
  • Lindsay, E. A., Mullen, C. J. Diversity and composition of soil fungal communities across land-use types in temperate forests. Soil Biol. and Biochem. 2020, 146, 107827. doi: 10.1016/j.soilbio.2020.107827.
  • Wardle, D. A., Jonsson, M., Berg, A. M. Soil microbial communities and their role in ecosystem processes: An overview. Soil Biol. and Biochem. 2005, 37(11), 1707-1715. doi: 10.1016/j.soilbio.2005.01.004.

  • Comment: Refine the guiding significance of the conclusion for forest management: In the conclusion section, more specifically elaborate on how the research results can provide guidance for forest management practices. For example, propose specific fungal community management strategies for different tree species.
  • Response: The conclusions have been expanded considerably to include additional guidance.

Round 2

Reviewer 1 Report

Comments and Suggestions for Authors

Authors answered my major concerns, I recommend it to be published.

Reviewer 3 Report

Comments and Suggestions for Authors

I think it can be accepeted now.